# Role and Potential of Artificial Intelligence in Biomarker Discovery and Development of Treatment Strategies for Amyotrophic Lateral Sclerosis

**DOI:** 10.3390/ijms26094346

**Published:** 2025-05-02

**Authors:** Yoshihiro Kitaoka, Toshihiro Uchihashi, So Kawata, Akira Nishiura, Toru Yamamoto, Shin-ichiro Hiraoka, Yusuke Yokota, Emiko Tanaka Isomura, Mikihiko Kogo, Susumu Tanaka, Igor Spigelman, Soju Seki

**Affiliations:** 1Laboratory of Neuropharmacology, Section of Biosystems and Function, School of Dentistry, University California, Los Angeles, 714 Tiverton, Los Angeles, CA 90095, USA; 2Department of Oral and Maxillofacial Surgery, Graduate School of Dentistry, The University of Osaka, Yamadaoka, Suita 565-0871, Japan; 3Division of Dental Anesthesiology, Faculty of Dentistry, Graduate School of Medicine and Dental Sciences, Niigata University, Niigata 951-8514, Japan

**Keywords:** neurodegenerative diseases, amyotrophic lateral sclerosis, artificial intelligence in medicine, machine learning algorithms, biomarkers, clinical trials, proteomics, neuroimaging, generative adversarial networks, disease biomarkers

## Abstract

Neurodegenerative diseases, including amyotrophic lateral sclerosis (ALS), present significant challenges owing to their complex pathologies and a lack of curative treatments. Early detection and reliable biomarkers are critical but remain elusive. Artificial intelligence (AI) has emerged as a transformative tool, enabling advancements in biomarker discovery, diagnostic accuracy, and therapeutic development. From optimizing clinical-trial designs to leveraging omics and neuroimaging data, AI facilitates understanding of disease and treatment innovation. Notably, technologies such as AlphaFold and deep learning models have revolutionized proteomics and neuroimaging, offering unprecedented insights into ALS pathophysiology. This review highlights the intersection of AI and ALS, exploring the current state of progress and future therapeutic prospects.

## 1. Introduction

Neurodegenerative diseases pose a significant global health challenge and burden individuals, societies, and healthcare systems [1]. These disorders result from complex mechanisms, such as neurodegeneration, inflammation, and ischemia, and present with diverse symptoms, including cognitive, motor, and sensory impairments [2,3]. ALS is characterized by the progressive degeneration of upper and lower motor neurons, which leads to severe muscle weakness and death within 3–5 years, primarily from respiratory failure [4,5,6]. In conditions such as ALS, for which curative treatments are lacking, early detection and appropriate treatment are essential to slowing disease progression and maintaining quality of life [4]. However, definitive diagnosis remains difficult because clinical assessments often rely on non-specific symptoms and imaging findings [7], underscoring the need for reliable biomarkers to enable early, precise diagnosis and monitoring [4,8].

Artificial intelligence (AI) is pivotal in clinical trials [9]. Recently, trials have become more complex and costly, and machine learning has been used to optimize trial design and participant selection [10,11]. AI enables rapid, comprehensive data analysis, enhancing predictions of treatment efficacy and accelerating drug development. AI models can predict pharmacokinetic parameters, simulate drug distribution and clearance, and optimize dosing regimens and administration routes. Consequently, they reduce the need for animal experiments and human trials, offering a cost-effective approach [12,13]. Moreover, next-generation sequencing, advanced computation, and novel algorithms have created new opportunities to predict patient outcomes and prioritize trials. Currently, AI evaluates disease progression and carries out stage classification in large-scale neurodegenerative trials and is expected to contribute significantly to the development of new therapeutic agents [14,15,16] (Figure 1).

Recent advancements in genomics, metabolomics, proteomics, and broader omics research have enhanced integrated analyses for biomarker discovery [17]. Omics imaging combines omics data (including genomics, transcriptomics, and proteomics) with structural, functional, and molecular imaging data to facilitate this search [17]. Advances in genetic testing have revealed that most patients with ALS have identifiable genetic causes of disease [18]. However, the complexity of these datasets necessitates the use of AI, with machine and deep learning techniques that extract patterns from large-scale data to identify reliable biomarkers for prediction of disease course, diagnosis, and pathogenesis [19,20,21,22]. AI has already automated and enhanced diagnostics based on medical imaging and shows promise for the early detection of neurodegenerative diseases [23,24,25]. Moreover, by clarifying disease pathogenesis, AI aids in identifying new therapeutic targets. In omics data analysis, established genomic methods such as cluster analysis and machine learning have been widely used; however, generative adversarial networks (GANs), a class of deep learning models, have more efficiently identified disease-related gene-expression patterns and molecular networks [23,26,27,28].

AI also has uses in proteomics: Jumper et al. [29], who developed the AI-based protein structure-prediction model “AlphaFold”, were awarded the Nobel Prize in Chemistry in 2024. This technology has significantly enhanced the efficiency of proteomics research, yielding results that far exceed the limitations of traditional structural-analysis methods. The potential applications of AlphaFold extend beyond cancer research, holding substantial promise for the development of treatments for neurodegenerative diseases and providing new insights into their underlying pathologies [30]. In data analysis, the volume of data is increasing, and deep learning models have been developed to extract highly abstract features and patterns from large datasets. This technology has been particularly prominent in cancer research for the identification of novel biomarkers through proteomic analysis [31,32,33].

AI is also being employed in neuroimaging analysis, particularly in the classification of magnetic resonance imaging (MRI) images from patients with ALS [34,35]. Machine learning algorithms are facilitating the identification of disease subtypes, enhancing diagnostic accuracy, and contributing to advancements in neuroimaging technology. Deep learning models, including Convolutional Neural Networks (CNNs) and Support Vector Machines, are also widely used in the analysis of MRI images from patients with neurodegenerative diseases [23,24,25]. Additionally, in connectomics, a field that aims to map and understand the neural circuits throughout the brain, FusionNet, a deep learning-based image-processing model, is used to analyze neuronal structures, such as cell membranes and nuclei, allowing the detection of abnormalities in disease-specific neural networks and thus further elucidating the underlying pathological mechanisms [36,37]. Through these methods, AI promotes a multifaceted understanding of neurodegenerative diseases, from molecular insights to circuit-level pathophysiology.

Recently, reviews have examined the intersection of ALS and AI [38,39,40,41,42,43,44,45] (Table 1). We reviewed recent advances in ALS biomarkers and diagnostics, explored how AI can clarify the pathophysiology of neurodegenerative diseases, and examined AI’s potential for developing novel ALS treatments. Finally, we provide an outlook on future AI-driven therapeutics for ALS.

## 2. Methodology

This review article is not a systematic review, but was written with reference to the PRISMA guidelines. PubMed was used to search for two keywords, one each for “ALS” or “amyotrophic lateral sclerosis” and “AI” or “artificial intelligence”. In total, 342 papers were extracted.

The inclusion criterion was the actual use of AI in the diagnosis, examination, or treatment of ALS or in research on ALS. As part of the selection process, all review articles were included and summarized in a table. The accompanying references were added to explain in detail the application of AI to ALS that appeared in the selected papers.

## 3. AI for Screening and Diagnosis of ALS

### 3.1. Diagnosis of Neurodegenerative Diseases

To understand the pathophysiology of complex neurological diseases and identify early biological markers for diagnosis and treatment efficacy, neuroimaging techniques (MRI, functional MRI, and positron emission tomography [PET]) have been extensively used [46,47,48,49]. The diagnosis of ALS involves a comprehensive evaluation, including patient history, physical examination, electromyography (EMG), evaluation of nerve-conduction velocity, and MRI. Additionally, electrophysiological tests—nerve-conduction studies and somatosensory evoked potentials—and the Revised ALS Functional Rating Scale (ALSFRS-R), which gauges severity, are employed. This complex diagnosis requires expertise in both clinical and electrophysiological testing and captures changes in the motor cortex and corticospinal tract [35,49,50,51]. PET is a powerful molecular imaging tool that assesses the distribution and binding of radiolabeled compounds to biologically relevant molecules [51]. Furthermore, neurophysiological testing, notably EMG, critically evaluates peripheral nerve function and sensory pathways to quantify the extent of motor neuron damage, an essential component of ALS diagnosis [52,53,54].

Cerebrospinal fluid (CSF), which surrounds the central nervous system (CNS), is rich in brain-specific proteins that can be preferentially detected over blood-derived proteins. Advanced biochemical and molecular techniques allow for the identification of specific proteins and RNA molecules in both blood and CSF, and these can serve as reliable diagnostic markers for various neurodegenerative conditions [55,56].

### 3.2. Integration and Impact of AI in ALS Diagnosis

In recent years, integration of AI and machine learning into diagnostics has advanced significantly. AI-driven image analysis now surpasses manual analysis in accuracy and reproducibility [6,57] and addresses challenges such as scarcity of labeled data and data imbalance [6,57]. Ker et al. reported that GANs facilitate the generation of synthetic data, thereby augmenting existing datasets and enhancing performance [57]. Future AI applications in medical diagnostics may include advancing radiation genetics, improv patient safety, and automating complex diagnostic tasks. However, as AI evolves in medicine, ethical concerns and algorithm transparency remain critical to the work [6,57]. Papi et al. demonstrated that blood tests using the biomolecular corona around nanoparticles exhibit high specificity and sensitivity in reducing mortality among early-stage cancer patients, suggesting that similar diagnostics could be developed for neurodegenerative diseases by identifying the nano-accumulation of plasma proteins correlated with disease states [20]. In large biomolecular corona datasets, AI can detect disease-specific patterns, enabling the early and precise identification of various diseases.

AI technologies are increasingly used to detect a wide range of diseases and improve diagnostic processes across multiple disciplines, including the diagnosis of neurodegenerative diseases such as ALS [19,20]. Additionally, more accurate diagnoses can be achieved by integrating multiple types of data, such as images, genomic data, and clinical information, for comprehensive analysis. In cancer research, Kosvyra et al. observed a growing reliance on data-driven AI methods to improve decision-making accuracy and efficiency. In Europe, the INCISIVE project is underway; this project utilizes cancer Digital Imaging and Communications in Medicine images to address challenges related to data availability and facilitate the widespread adoption of AI solutions in medical imaging [58]. Active research in both AI and precision medicine by Johnson et al. suggests a future in which healthcare professionals and consumers will benefit from highly personalized diagnostic and treatment information. The synergy between AI and precision medicine is expected to revolutionize healthcare by enabling early disease detection and prevention, thereby reducing the overall disease burden and associated healthcare costs [59].

AI is finding uses in the diagnosis and treatment of various diseases; these methods are also being explored in work that to facilitate the diagnosis of ALS.

Huber et al. utilized proteomic data derived from induced pluripotent stem cell (iPSC)-derived motor neurons from patients with ALS, which were available through the AnswerALS consortium, to build a statistical classification machine learning model using ridge regression. This model distinguished between patients with ALS and controls with high accuracy [60]. The machine learning model identified 10 diagnostic biomarker proteins that could distinguish patients with ALS from healthy controls with notable sensitivity and precision [60]. Imamura et al. employed iPSCs to construct an artificial intelligence-based ALS-prediction model, analyzing images of spinal motor neurons from both healthy control participants and patients with ALS using a CNN [61]. The deep learning model generated in this study was able to classify ALS and healthy controls with high accuracy. This report suggests that this predictive model, using iPSC technology and deep learning algorithms, holds promise for supporting diagnosis and guiding potential treatments for ALS through future prospective studies [61].

We believe that the integration of various AI-assisted technologies will lead to more precise ALS diagnoses in the future, facilitating the selection of personalized treatments [59,62,63,64].

Segura et al., a research group based in Spain, reported a study that utilized natural language processing AI tools to automatically extract clinical information from a large patient population. They found that major ALS symptoms, such as dyspnea, weakness, dysarthria, fasciculation, and dysphagia, often appeared before ALS was officially documented in a patient’s electronic medical record. The failure to recognize these early symptoms resulted in a significant diagnostic delay, averaging 11 months [65]. Additionally, only one-quarter of patients were referred to a neurologist 1 year before their ALS diagnosis, with dyspnea being particularly strongly associated with delayed referral and diagnosis [65]. Bede et al. developed an observer-independent multiclass (three-way) classification protocol for classifying multi-parameter imaging data from a large cohort of participants consisting of 214 patients with ALS, 127 healthy controls, and 37 disease controls [66]. Using a multilayer perceptron model within an artificial neural network framework, they found that white-matter indices were significantly more associated with ALS than were gray-matter indices [66].

Recently, review papers on the use of AI in ALS have begun to emerge (Table 1). In a systematic review of ALS and AI by Umar et al., a meta-analysis was conducted that involved extracting data from 34 studies. The pooled sensitivity and specificity of the AI model were 94.3% and 98.9%, respectively, surpassing those of conventional ALS diagnostic methods [38]. They concluded that AI could play an important role in the screening and diagnosis of ALS because of its high sensitivity and specificity. However, concerns regarding the quality of the evidence in the literature remain [38]. In a systematic review by Tavazzi et al., it was stated that AI is valuable for the stratification and prediction of ALS progression [39]. Although the superiority of deep learning-based models for prediction of ALS progression over conventional methods has not yet been conclusively demonstrated, there remains significant potential for applying AI in patient stratification. In a review by de Jonge et al., the potential use of AI in the interpretation of clinical needle electromyography signals necessary for ALS diagnosis was discussed. Although many studies have reported highly accurate classification models, the authors cautioned that issues such as bias and overfitting remain concerns and that current models are insufficient for clinical implementation [22]. These issues will be addressed in detail in a later section of this article.

## 4. Application of AI in ALS Biomarkers

### 4.1. Role of Biomarkers in Neurodegenerative Diseases

Reliable CNS biomarkers are crucial for diagnosing and treating neurodegenerative diseases [67,68]. Although tissue biopsy and neuroimaging are used to evaluate CNS tumors, they offer incomplete and sometimes ambiguous data. Consequently, circulating biomarkers—including circulating tumor cells, circulating tumor DNA, and extracellular vesicles (EVs)—are emerging as promising real-time tools for liquid biopsy. They can be used to monitor tumor burden, disease progression, and treatment response and provide dynamic genetic profiles for adaptive management. In neurodegenerative diseases, recent CSF biomarker development has established clinical diagnostic tools that reflect CNS pathology and effectively monitor proteins, gene expression, and metabolites [68,69]. In ALS, EV cargo from patient blood samples has been investigated for biomarker development, elucidation of pathological mechanisms, and therapeutic applications [31,70,71,72,73,74,75,76,77,78,79,80,81] (Table 2).

### 4.2. Role of Biomarkers in ALS

Biomarker research in ALS deepens our understanding of its pathogenesis and pathology [68,76]. Neuropathologically, ALS is associated with degeneration and death of upper and lower motor neurons; however, recent studies revealed the involvement of sensory neurons. Seki et al., 2019 reported an abnormality in the trigeminal nucleus, the primary controller of mastication, in infants with the SOD1G93A mutation [5,6,82,83,84]. In a review, Seki et al. noted that primary sensory (dorsal root ganglion [DRG]) neurons accumulate misfolded proteins and exhibit mitochondrial abnormalities, reduced high-voltage-activated Ca^2+^ currents, and splice variants of peripherin, a biomarker of axonal damage [5]. Skeletal muscle also can serve as a potential biomarker for ALS diagnosis. Kawata et al. observed that progression of muscle atrophy varies significantly across ALS-affected skeletal regions, with the masseter muscle in ALS mice (SOD1G93A) resisting atrophy until the end stage [85]. ALS and frontotemporal dementia (FTD) are devastating neurodegenerative diseases characterized by progressive motor=neuron loss and cognitive decline, respectively [86]. The discovery of hexanucleotide repeat expansion mutations in the C9orf72 gene has spurred intense research into the pathogenesis of these diseases and the development of targeted therapies [87]. Additionally, four novel potential genetic biomarkers for ALS have been identified in human bone marrow mesenchymal stem cells derived from patients: TAR DNA-binding protein 43 (TDP-43), secretory leukocyte protease inhibitor, CyFIP2, and RbBP9. Abnormal aggregation of TDP-43 is a major pathological feature of ALS [76]. TDP-43, a key component of cytoplasmic ubiquitinated protein inclusions in neurons and glial cells in both sporadic and familial ALS, shows significant nuclear depletion in ALS [88]. In most cases of sporadic ALS and many cases of frontotemporal lobar degeneration, TDP-43-positive neuronal cytoplasmic inclusions, which are hyperphosphorylated and ubiquitinated, occur in the brain and spinal cord [89]. In ALS pathology, TDP-43 aggregation is a major feature, and proteomic analysis has identified proteins that interact with TDP-43 [31,32,76]. Recent advances in proteomics have enabled the detection of protein mutations in neurodegenerative diseases.

Novel RNA biomarkers, such as plasma cell-free microRNA (miR-181), have also been developed for ALS. It has been shown that microRNA (miRNA) is an endogenous non-coding RNA essential for the survival of motor neurons and that miR-181 is downregulated in post-mortem ALS motor neurons [73,90]. Moreover, mutations associated with ALS have been reported to affect the activity of the DICER complex and disrupt miRNA biosynthesis, suggesting a role in altering neuronal integrity. This disruption underscores the potential of miRNAs as therapeutic targets in ALS [73,90]. The neurofilament light chain protein (NFL) is an inflammatory cytokine and has been reported to be the first blood biomarker that could be used to predict the progression rate of ALS and stratify patients based on their expected survival times [73,91]. Recent studies have demonstrated that combining NFL with miR-181 enhances the accuracy of prognostic predictions, allowing for better patient stratification and more precise treatment planning [73,74,75] (Figure 2).

Schematic showing neuronal and glial abnormalities in ALS. The figure illustrates motor and sensory neuron involvement, including hyperexcitability, axonopathy, and glial dysfunction, with associated gene mutations. Figure created with BioRender.com (accessed on 20 February 2025).

Biomarkers are also crucial in drug development. Because biomarker research began in the early 2000s, many studies have been published; however, the number of biomarkers currently used in clinical practice remains relatively small. This is primarily due to the limited number of participants in biomarker studies, which results in low statistical power, as well as the lack of robust validation and standardization of the methods used. The discovery and development of reliable biomarkers has the potential to significantly advance patient stratification and personalized medicine by providing insights into the molecular mechanisms of diseases [50,92,93]. These biomarkers are vital for selecting appropriate clinical-trial participants and monitoring treatment efficacy [50,67,78,92,93]. While numerous drugs have entered clinical research for ALS, many have failed to demonstrate definitive efficacy [23].

In addition, the clinical heterogeneity of ALS arises from the variable combination of upper- and lower-motor-neuron signs, which complicates diagnosis, tracking of disease progression, and monitoring of treatment outcomes, particularly in clinical trials. Therefore, it is crucial to assess both upper- and lower-motor-neuron dysfunctions. Neurophysiological biomarkers that electrophysiologically evaluate motor neurons are actively being developed. According to Huynh et al., transcranial magnetic stimulation (TMS) can be employed by placing a TMS coil on the scalp of a patient with ALS to activate the primary motor cortex, which induces hyperpolarization of the postsynaptic neurons. It has been reported that TMS, including single-pulse TMS and paired-pulse TMS, is particularly useful for monitoring ALS progression [94].

### 4.3. Research on the Development of Biomarkers for ALS Using AI

The identification and validation of new biomarkers for ALS remain critical for advancing diagnostic accuracy and therapeutic approaches. The combination of multiple biomarkers has the potential to significantly improve both the diagnosis and prognosis of ALS [73,76,95,96,97,98]. One promising approach for identifying novel biomarkers is omics imaging, an emerging interdisciplinary field that combines biomedical imaging with omics data (such as genomics, proteomics, and metabolomics). Omics imaging seeks to uncover complex genotype–phenotype relationships by integrating data from multiple sources, thus providing deeper insights into disease mechanisms, onset, and progression, while also identifying new diagnostic and prognostic biomarkers [17]. Hariri and Weinberger have highlighted the potential of integrating functional neuroimaging with functional genomics, predicting that this combination will yield a more comprehensive understanding of human behavior and the pathological states of the brain [17]. Similarly, Xu et al. explored genetic diseases by merging information from genome-wide association studies with large-scale imaging datasets, facilitating a more thorough elucidation of disease mechanisms [17]. Katrib et al. reviewed the increased availability of RNA- sequencing data and noted its growing importance in the integration of transcriptomic data into disease research [17]. Ongoing advancements in AI-driven large-scale analysis can similarly be applied to ALS research, offering promising avenues for identifying new biomarkers, tracking disease progression, and ultimately improving diagnostic and therapeutic strategies.

AI has also been applied to image analysis for patients with ALS, providing valuable insights into disease progression. Behler et al. reported that diffusion tensor imaging (DTI) can be used to image patients with ALS in vivo, capitalizing on the fact that the distribution of pTDP-43 follows a continuous pattern across four distinct neuropathological stages in the CNS [41]. This approach facilitates in vivo tracking of pTDP-43 distribution, which has become a hallmark of ALS pathology. Furthermore, DTI enables a more nuanced understanding of white-matter degeneration in ALS, particularly of the degeneration of the upper motor neurons. Behler et al. employed a combination of DTI, eye-movement recordings, and cognitive testing using the ECAS in a cohort of 245 patients with ALS. Using hierarchical clustering, they identified four distinct clusters, offering a deeper understanding of the clinical heterogeneity of ALS and suggesting the utility of AI for stratifying patients with ALS based on disease characteristics [41]. Based on FA values, cognitive assessments, and eye-movement analysis, each cluster provided an indicator of disease progression. AI-based analysis confirmed correlations between structural and functional brain changes, enabling in vivo stage classification at the individual level and quantitative mapping of disease progression [41].

In clinical practice, advancements in sensor technology and data-processing capabilities have led to more sensitive and accurate measurements, enabling the earlier detection of pathological changes in patients with ALS. AI-assisted imaging techniques have also significantly enhanced image processing and detection [6,78]. One notable example is the threshold-tracking TMS technique used in paired-pulse TMS, which allows for the examination of short-term intracortical inhibition and intracortical facilitation in patients with ALS. By applying subthreshold conditioning stimuli at predetermined time intervals before suprathreshold test stimuli, this technique provides insights into motor-cortex dysfunction, which is often observed in patients with ALS [94]. Neurophysiological biomarkers facilitate non-invasive measurements, making ALS diagnosis simpler and more accurate. Combining electrophysiological and behavioral physiological assessments offers substantial benefits, particularly when these assessments are integrated with sensor technologies. For instance, Kitaoka et al. developed an AI model using the Single Shot Multibox Detector, a deep learning-based object-recognition algorithm, to track mouth-opening and -closing movements in ALS mouse models (SOD1G93A). This model demonstrated a correlation between the extension of the chewing cycle and weight loss in ALS mice, highlighting the potential of behavioral biomarkers as indicators of disease progression [6]. Nakamori et al. examined the utility of swallowing-sound assessment in 24 patients with ALS using an electronic stethoscope, with a subsequent analysis performed by an AI-based tool. Their findings revealed a strong correlation between the results of this method and traditional swallowing- assessment indices, demonstrating its potential as a novel diagnostic approach [99]. Pancotti et al. developed a predictive model of ALS progression using data from the Pooled Resource Open-Access ALS Clinical Trials (PRO-ACT) repository, one of the most comprehensive open-access data resources on ALS. By applying deep learning techniques, they showed that the model accurately predicted the ALSFRS progression slope in patients with ALS [100].

Detailed monitoring of disease progression using such biomarkers can enhance the assessment of treatment efficacy and contribute to the development of new therapeutic strategies [6,17,94,101]. Ultimately, the integration of biomarkers for early diagnosis and personalized treatment, augmented by AI, is expected to improve the prognosis and quality of life of patients with neurodegenerative diseases, including ALS [10,11,12,19,20,23,24,25,102,103,104,105,106,107,108] (Table 3).

## 5. New Approaches to ALS Treatment

### 5.1. Development of Future ALS Treatment Methods

Riluzole, a glutamate neurotransmitter-release inhibitor, and edaravone, a free- radical scavenger that prevents cell damage and slows early ALS progression, are currently approved for ALS patients. However, these treatments yield only modest survival benefits and have limited effects [109,110]. Therefore, innovative therapeutic approaches offering substantial and fundamental cures for ALS are urgently needed.

Stem-cell therapy has recently gained significant attention for its potential to target multiple disease mechanisms and slow ALS progression. Notably, it provides nutritional and immunomodulatory support while promoting motor-neuron regeneration. Neural and mesenchymal stem cells have differentiated into motor neurons in vitro, and their transplantation in ALS animal models has improved symptoms and motor function [110,111]. Additionally, iPSC therapy shows promise. Pun et al. utilized PandaOmics, an AI-driven target-discovery platform, to analyze gene-expression profiles from CNS samples (237 cases, 91 controls) and iPSC-derived motor neurons from the AnswerALS initiative (135 cases, 31 controls), identifying 17 highly reliable and 11 novel therapeutic targets, thereby advancing iPSC-based therapies for ALS treatment [112]. A major challenge in the development of CNS-targeted drugs is the inability of many compounds to cross the blood–brain barrier (BBB). To address this, nanoparticle-based drug-delivery systems have gained attention as non-invasive, safe methods for targeted delivery of drugs to the brain, effectively bypassing the BBB and offering a promising solution for CNS disorders, including ALS [113].

Gene therapy represents another promising treatment strategy for ALS. Vectors used in gene therapy include the oncolytic virus HSV-1, which is mainly used in cancer treatment in clinical settings [114,115] and has already been used in clinical trials; adeno-associated virus (AAV) vectors; and EIAV (equine infectious anemia virus) vectors, which have already been used in clinical trials for neurodegenerative disease [116]. Founta et al. introduced a novel gene-selection methodology utilizing semi-automatic preprocessing with SES, a causal feature-selection algorithm specifically designed for high-dimensional datasets including a limited number of patients with ALS [117]. One approach to treating neurodegenerative diseases involves RNA degradation via antisense oligonucleotides, whereas proteins can be supplemented using AAV vectors to address motor dysfunction [118]. For instance, AAV-miRSOD1, which silences the SOD1 gene, has been shown to suppress ALS onset in SOD1G93A mice, a common mouse model for ALS [119].

IBM Watson^®^ is an advanced platform for AI and data analytics that facilitates the development and deployment of machine learning models and generative AI. It consists of three core components and an AI assistant designed to expand and accelerate AI applications using reliable data [120]. One of its applications, Watson for Oncology, supports cancer treatment decision-making [116]. Bakkar et al. leveraged IBM Watson^®^ to analyze RBPs in the genome, identifying novel RBPs associated with ALS. They ranked these RBPs based on semantic similarity from the published literature and validated their findings in the tissues and stem cells of patients with ALS, identifying five new RBPs (hnRNPU, Syncrip, RBMS3, Caprin-1, and NUPL2) linked to ALS [120]. These results highlight the potential for AI to accelerate neurological disease research. Additionally, it has been demonstrated that an inhibitor of Src/c-Abl kinase promotes autophagy and protects motor neurons derived from ALS iPS cells against degeneration. Various molecularly targeted drugs that modulate these pathways are currently under development [61]. Boyce et al. investigated the causes of ALS by interviewing 3061 patients with ALS and analyzing their responses using AI-driven methods. Their findings suggest that AI has the potential to overcome barriers to understanding the patient experience by integrating first-hand data from patients with ALS [40]. Furthermore, while traditional ALS research has focused on motor neurons, Seki et al. have proposed the novel possibility of targeting primary sensory neurons, such as those in the DRG, which receive muscle-spindle information, as part of ALS gene therapy [5,121]. In addition, Seki et al., 2023 mentioned the possibility that neuropeptide Y (NPY) and serotonin may be used as therapeutic agents targeting the primary sensory neurons of ALS [121,122].

### 5.2. Development of Therapeutic Strategies Using AI

AI is pivotal in developing treatments for neurodegenerative diseases (Table 2). In drug discovery, AI identifies new uses for existing drugs (drug repurposing) and designs novel compounds. Recent advancements underline AI’s role in drug repurposing for COVID-19 treatments [123]. For ALS, an AI-based platform has been created for personalized drug discovery and identification of molecular biomarkers; it predicts therapeutic properties using a comprehensive database of transcriptomic and transcriptional response data from human cell lines exposed to various molecules [12,13,104,123,124,125,126]. Furthermore, AI has been employed in rehabilitation research to implement a self-service telemedicine system that remotely analyzes voice tests to detect dysarthria progression [127,128]. Notably, AI has served as a communication tool for patients with ALS via a brain–computer interface using an intracortical voice prosthesis. After installation in a patient with ALS, the prosthesis achieved 99.6% accuracy with a 50-word vocabulary just 25 days post-operation, maintained 97.5% accuracy over 8.4 months, and enabled communication at approximately 32 words per minute [129]. In addition to these clinical applications, AI-based protein-structure-prediction tools such as AlphaFold have opened promising avenues for target identification in drug development [29,30]. However, it should be noted that these approaches remain limited in the context of neurodegenerative diseases involving intrinsically disordered proteins (IDPs), such as TDP-43 in ALS, due to the dynamic and context-dependent nature of these proteins [130,131].

### 5.3. Progress in Personalized Medicine

Most neurodegenerative diseases are extremely complex, involving numerous interactions between genetic, environmental, and lifestyle factors. Given this diversity, personalized treatment tailored to specific conditions is crucial. Personalized medicine involves the use of an individual’s unique genomic information to optimize patient care. Because disease onset, progression, prognosis, and drug response vary among individuals, the primary goal is to effectively apply genomic insights to clinical practice. Genomic analysis helps identify genetic factors that influence disease susceptibility and pathology, which in turn informs stratified treatment approaches and gene-therapy strategies using nanomedicine [132,133]. AI also plays a critical role in realizing personalized medicine. By accurately predicting disease progression in individual patients based on genomic data and clinical information, AI assists in determining the most appropriate treatment methods and dosages for each patient’s specific disease trajectory [105,134,135]. McGown et al. reviewed the use of various high-throughput screening (HTS) platforms in ALS drug discovery, noting that the introduction of AI technologies could improve target selection and compound optimization in HTS, thus enhancing clinical trials by accounting for the genetic profiles of patients [42]. Marriott et al. employed machine learning to conduct hierarchical clustering, demonstrating that gene-expression data can effectively stratify patients with ALS into molecular and phenotypic subgroups [136]. Their findings suggest that distinct ALS pathologies are driven by different underlying mechanisms, which can be identified through specific gene-expression signatures, enabling the development of personalized treatments [136]. Integrative analysis of omics data, imaging, and clinical information is set to provide deeper insights into individual disease states, paving the way for tailored treatment strategies and advancing the development of targeted therapies [113,133]. This approach to personalized medicine is anticipated to significantly enhance disease prevention, early intervention, and patient outcomes.

### 5.4. Advantages and Limitations of AI Technology

AI-assisted diagnosis offers advantages owing to its advanced analytical capabilities, which enable the early detection of subtle changes and the identification of disease [6,58,62,63]. AI can reveal novel disease patterns in large datasets, thereby deepening our understanding of the pathology [19,20,21,58]. Furthermore, advances in visualization technologies allow the presentation of complex data in more intuitive formats, supporting clinicians in making accurate and reproducible diagnoses while reducing their workload [62,63,137].

However, several limitations persist. A major concern is the “black box” nature of AI decision-making, wherein the output derivation remains opaque [137]. As a response to concerns regarding the black-box nature of artificial intelligence, the adoption of Explainable AI (XAI) has garnered increasing attention. XAI aims to present the basis and reasoning process behind the outputs of machine learning models—particularly deep learning models—in a form that is comprehensible to humans. This approach has potential for improving algorithmic transparency and ensuring verifiability by end users [138,139]. In particular, in domains such as medicine and the life sciences, explainability in the decision-making process is increasingly emphasized from both legal and ethical perspectives. The implementation of XAI is therefore expected to enhance the acceptability and trustworthiness of AI systems in these fields [138,139].

Nevertheless, current XAI techniques are subject to several technical limitations. They are not universally applicable to all model types, and the explanations provided may not always offer accurate or consistent interpretations. Consequently, challenges remain in terms of their practical utility and reliability [138,139]. Other challenges include data biases, variability in AI-tool effectiveness across healthcare providers, and automation bias, which may lead clinicians to rely excessively on decision-support systems. Additionally, AI expertise is crucial for accurately evaluating test results. The integration of AI in medical-image interpretation requires effective collaboration between clinicians and AI algorithms. Although AI has demonstrated the potential to improve clinical performance, its impact on individual clinicians remains unclear and may even detrimentally affect diagnostic accuracy. Similar limitations have been noted in the field of structural biology. As outlined earlier, although AlphaFold has had a transformative impact on protein-structure prediction, it encounters substantial difficulties in accurately modeling intrinsically disordered proteins (IDPs) and membrane proteins. These classes of proteins either lack well-defined tertiary structures or require specific environmental contexts such as lipid membranes or ligand interactions in order to acquire their functional conformations. These context-dependent structural features remain challenging for current AI-based prediction algorithms, including AlphaFold, to fully capture [130,131]. Therefore, ensuring the accuracy of AI models and understanding their effects on clinical decision-making is critical [140,141,142].

### 5.5. Future Impact of AI on ALS Treatment Strategies

Further advancements in AI are anticipated to substantially change treatment strategies for neurodegenerative diseases. By integrating multi-omics data and connectomics, AI can elucidate the molecular pathology of diseases and the underlying neuronal structures and neural networks, paving the way for the realization of personalized treatments [36,37]. Additionally, AI-driven drug discovery has the potential to accelerate the development of disease-specific therapeutic agents. AI is also revolutionizing drug-delivery technologies, enabling the creation of targeted, personalized, and adaptive therapies. AI-based data analysis and optimization methods are expected to empower pharmaceutical researchers and medical professionals to enhance drug efficacy, reduce side effects, and improve patient outcomes [12,13]. Moreover, the growing use of wearable devices and mobile applications will facilitate the large-scale collection and analysis of real-world data, including neurophysiological biomarkers that can be recorded non-invasively, clinical characteristics of diseases, and demographic statistics. Notable examples include smart applications that use tactile and visual stimulus modalities to monitor tremors (involuntary movements) and cognitive impairments (e.g., memory loss or dementia) through AI analysis.

This innovative approach is likely to enhance treatment monitoring and provide real-time feedback to patients and clinicians [135,143,144,145]. Thus, AI technology is expected to revolutionize the entire treatment paradigm for neurodegenerative diseases, making a significant contribution to improving quality of life for each patient.

### 5.6. Issues in Translational Research Using AI

In recent years, new ALS treatments have been developed by integrating clinical data, genetic information, and physiological measurements to construct disease models using machine learning technology [145]. Critical to this approach is the use of AI in preclinical research, such as proteomic analysis, and its effective application in translational research [135]. Specifically, by merging extensive non-invasive data—including clinical characteristics, demographic information, and diagnostic biomarkers (neurophysiological biomarkers)—into databases, AI technology significantly enhances analytical efficiency for the clinical translation of new treatment modalities [78,145,146]. In clinical settings, where personalized treatment plans are developed based on the design of therapeutic strategies, AI-driven intelligence is expected to improve decision-making [135].

Despite its potential, translational research using AI faces challenges. Large-scale clinical trials remain essential for validating the efficacy and safety of novel treatments, especially for emerging therapies such as gene therapy, which require critical long-term safety assessments [120]. Moreover, machine learning and AI-generated models are often criticized as “black boxes” with unclear prediction rationales; this disadvantage complicates their interpretation and potentially impedes their clinical application [120,121]. Additionally, establishing manufacturing and supply systems, evaluating cost-effectiveness, and assessing ethical and social implications require multifaceted efforts. Effective collaboration among patient groups, regulatory bodies, healthcare professionals, and researchers is vital to address these concerns [147,148].

### 5.7. Future Prospects for AI Technology in ALS

The continued evolution of AI technology promises to optimize diagnosis and improve clinical outcomes by enhancing early disease detection and timely, appropriate treatment. Moreover, consolidating AI models, datasets, and algorithms into a comprehensive, widely accessible database, even in remote and resource-limited areas, would broaden access to advanced diagnostic and treatment capabilities, significantly impacting clinical practice and public health [62,63,137]. AI plays a critical role, particularly in diseases such as ALS, for which treatment options are scarce and outcomes often remain poor. Furthermore, AI is being integrated into the rehabilitation and care of patients with ALS to improve treatment efficiency and quality of life [6,149]. Additionally, a systems-biology approach integrating multi-layered data, such as genomics, proteomics, and imaging, holds the potential to deepen our understanding of disease mechanisms and guide therapeutic development [6,58]. AI’s capacity to analyze large, complex datasets and learn from them could support clinicians in making more informed, data-driven decisions, ultimately contributing to personalized medicine by providing patients with the most appropriate diagnosis and treatment based on extensive data [59,64,102,103]. Specifically, AI- and data-driven approaches are expected to advance the identification of novel biomarkers, including new drug targets and neurophysiological biomarkers, thus deepening our understanding of ALS pathophysiology and leading to more effective treatments [17,64,76,104,122,124,125,126]. Moreover, advances in brain-function monitoring and neurofeedback technologies could leverage brain plasticity to develop neuroprotective strategies and support functional recovery [106,107,108]. With progress in regenerative medicine, innovative therapies such as the transplantation of artificially constructed neural networks are becoming increasingly feasible [150]. In clinical settings, AI integration will be indispensable in enhancing diagnostic and treatment capabilities and enabling the full utilization of emerging therapeutic approaches [135] (Figure 3).

Schematic representation of an AI-integrated workflow for ALS diagnosis and treatment. AI-assisted imaging analysis supports radiological interpretation, diagnosis, and personalized care, enhancing clinical decision-making and patient outcomes. Figure created with BioRender.com (accessed on 20 February 2025).

## 6. Conclusions

ALS remains a challenging neurodegenerative disease with limited therapeutic options and modest benefits from currently available treatments. However, recent advances in stem-cell therapy, gene therapy, nanomedicine, and AI offer promising new directions for personalized interventions. AI, in particular, is transforming ALS research by accelerating drug discovery, identifying biomarkers, and enabling precision medicine through integrated data analysis. New platforms highlight AI’s capacity to uncover novel therapeutic targets, and AI-driven communication tools enhance patients’ quality of life. Despite these advances, challenges such as a lack of transparency surrounding AI models, ethical concerns, and the need for validation studies remain. Therefore, the integration of AI with multi-omics and regenerative medicine has the potential to revolutionize ALS treatment. Achieving this will require collaborative efforts to ensure equitable access and meaningful clinical translation. AI’s role in ALS research is not just innovative; it is essential for advancing truly patient-centered care.

## Figures and Tables

**Figure 1 ijms-26-04346-f001:**
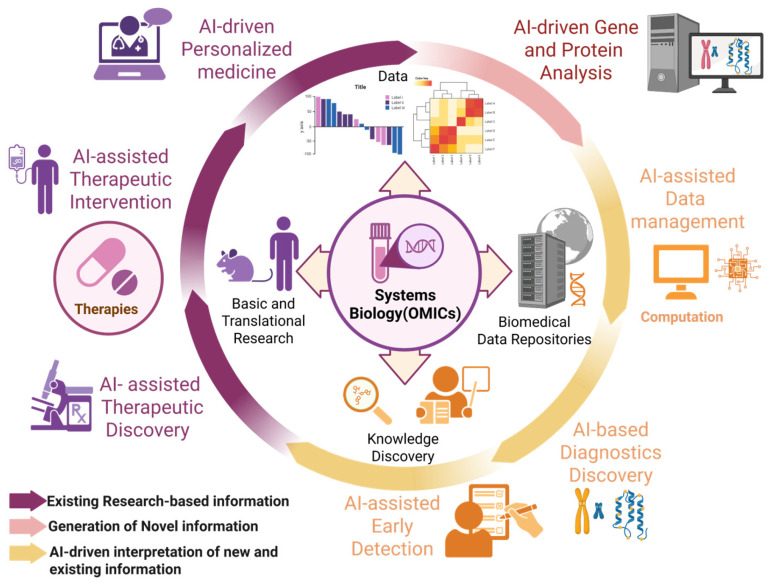
Role and potential of artificial intelligence for applications in ALS research and therapeutics. Conceptual diagram of the key domains in which artificial intelligence (AI) contributes to amyotrophic lateral sclerosis (ALS) research and care. Figure created with BioRender.com (accessed on 20 February 2025).

**Figure 2 ijms-26-04346-f002:**
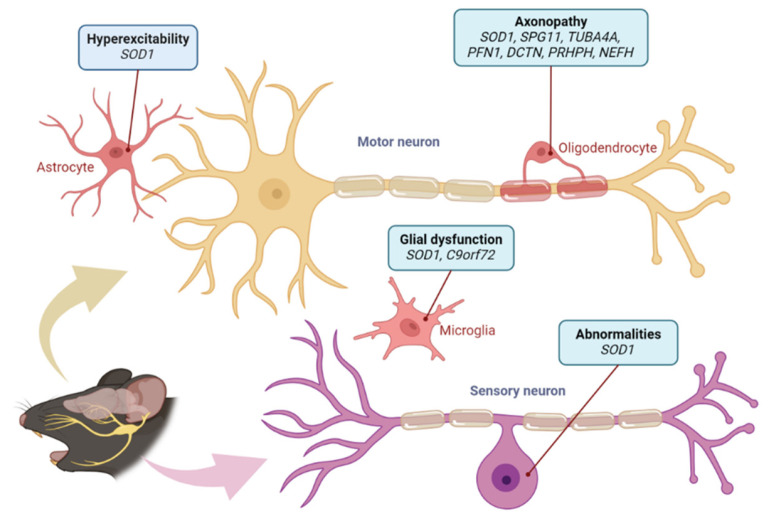
Abnormalities in neurons in ALS.

**Figure 3 ijms-26-04346-f003:**
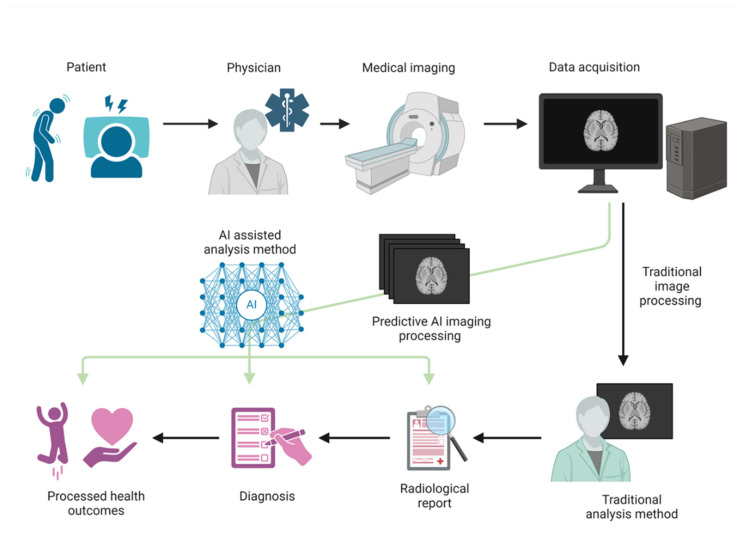
ALS treatment using AI assistance versus traditional methods.

**Table 1 ijms-26-04346-t001:** Trends in review articles on AI in ALS research.

Year	Authors	Title	Journal	Focus	Reference
2018	McGown & Stopford	High-Throughput Drug Screens (HTDS) or ALS Drug Discovery	Expert Opinion on Drug Discovery	Overview of HTDS methods and artificial intelligence (AI) applications in ALS drug discovery.	[42]
2020	Pinto et al.	New Technologies and Amyotrophic Lateral Sclerosis	Journal of Neurological Sciences	AI, telemedicine and assistive technologies accelerated by COVID-19.	[43]
2021	Fernandes et al.	Biomedical Signals and Machine Learning (ML) in Amyotrophic Lateral Sclerosis	BioMed Eng Online	ML applications in ALS diagnosis, communication, and survival prediction.	[44]
2021	Cooper-Knock et al.	Advances in the Genetic Classification of Amyotrophic Lateral Sclerosis	Current Opinion in Neurology	Genetic classification and ML models for understanding ALS.	[45]
2022	Behler et al.	Diffusion Tensor Imaging in ALS: Machine Learning for Biomarker Development	International Journal of Molecular Sciences	Use of diffusion tensor imaging (DTI) and ML for ALS biomarker discovery and stratification.	[41]
2023	Tavazzi et al.	AI and Statistical Methods for Stratification and Prediction of ALS	AI in Medicine	AI methods for stratification and prediction of ALS progression.	[39]
2024	Boyce et al.	What Do You Think Caused Your ALS?	ALS and Frontotemporal Degeneration	AI and qualitative methods to analyze patient-reported causes of ALS.	[40]
2024	Umar et al.	AI for Screening and Diagnosis of ALS	ALS and Frontotemporal Degeneration	Meta-analysis of AI tools for ALS screening and diagnosis.	[38]

**Table 2 ijms-26-04346-t002:** Overview of biomarker types and clinical applications in ALS.

Biomarker Type	Examples	Diagnostic Relevance	Associated Diseases	Details and Clinical Applications	References
Protein Biomarkers	Neurofilament light chain (NFL)	Indicator of axonal damage, correlates with disease severity	ALS, Alzheimer’s	Used in CSF and blood tests; prognostic marker for disease progression	[74,75,77]
Genetic Markers	TDP-43	Linked to neuronal degeneration, found in cytoplasmic inclusions	ALS, Frontotemporal dementia	Identifies TDP-43 proteinopathies; aids in differential diagnosis	[31,76]
	Mutations in SOD1	Common genetic cause of familial ALS	ALS	Screening in at-risk populations; genetic counseling	[70,71]
	C9ORF72 expansions	Most common genetic variation in familial ALS and FTD	ALS, FTD	Helps in confirming familial cases; guides prognosis and management	[72]
	TDP-43 mutations	Implicated in ALS pathology, affects RNA processing	ALS	Useful for familial ALS cases; potential targets for therapy	[76]
Molecular Biomarkers	Plasma cell-free miRNA	Non-invasive markers that reflect gene expression changes	ALS	Potential for early diagnosis and monitoring of disease progression	[73]
	The cargo content of extracellular vesicles (EVs)		ALS		[79,80,81]
Electrophysiological Biomarkers	Motor Unit Number Estimation (MUNE)	Quantifies the number of functional motor units	ALS	Assesses disease progression and response to treatment in ALS	[78]

**Table 3 ijms-26-04346-t003:** Applications of AI in ALS management.

AI Application	Techniques Used	Description and Use Cases	Impact and Clinical Relevance	Associated Diseases	References
Diagnostic Imaging	Deep Learning, Convolutional Neural Networks	AI algorithms analyze MRI, PET scans to detect and quantify pathological changes	Enhances accuracy and speed of diagnosis	ALS, Alzheimer’s, Parkinson’s	[23,24,25]
		Automated measurement of brain atrophy and detection of specific protein accumulations	Provides early-detection capabilities		
Drug Discovery	Machine Learning, Network Analysis	Identification of new drug targets and repurposing of existing drugs	Speeds up drug-discovery process, reduces costs	ALS, Alzheimer’s, Parkinson’s	[12,104]
		AI-driven simulations predict drug interactions and effectiveness	Improves safety and efficacy of new drugs		
Clinical Trials	Deep Learning, Predictive Analytics	Optimization of clinical-trial design and participant selection	Increases efficiency and efficacy of trials	Neurodegenerative diseases	[10,11]
		Real-time data analysis predicts treatment outcomes	Facilitates faster regulatory approvals		
Personalized Medicine	Machine Learning, Genomic Data Analysis	Customization of treatment plans based on patient genetic profiles	Enhances treatment effectiveness and reduces adverse effects	Neurodegenerative diseases	[102,103,105]
		AI models predict disease progression and treatment responses	Allows timely adjustments to therapy		
Neurorehabilitation	AI-driven Robotics, Neurofeedback	AI algorithms control robotic devices for physical therapy	Improves motor function and recovery rates	Stroke, ALS, Parkinson’s	[106,107,108]
		Neurofeedback techniques train patients to modify brain activity	Enhances cognitive rehabilitation		
Predictive Analytics	Machine Learning, Big Data Analysis	Analysis of large-scale health data to predict disease trends	Aids in public-health planning and resource allocation	Neurodegenerative diseases	[19,20]

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
