# Peer review of "Role and Potential of Artificial Intelligence in Biomarker Discovery and Development of Treatment Strategies for Amyotrophic Lateral Sclerosis"

_ijms, 2025, doi:10.3390/ijms26094346_

Round 1

Reviewer 1 Report

Comments and Suggestions for Authors

This paper presents a thorough and comprehensive overview of how artificial intelligence (AI) is changing the search for biomarkers, diagnostics, and therapy development for amyotrophic lateral sclerosis (ALS). Drawing from many artificial intelligence fields—machine learning, deep learning, and generative models—it synthesizes much research and applies it to proteomics, genomics, neuroimaging, and clinical practice. The logical organization includes tables summarizing AI applications in ALS (for example, diagnostic biomarkers therapeutic techniques) and previous reviews. It covers diagnosis, the creation of biomarkers, new therapies, and personalized medicine. The authors also included real-world data from translational studies and preclinical models, like those using SOD1G93A ALS mice, AlphaFold for proteomics, and clinical projects like the AnswerALS database. The authors also included real-world data from preclinical models, like those using SOD1G93A ALS mice. The generally well-referenced, current, and theoretically rich paper is a potentially beneficial tool for researchers and doctors in neurology and AI-driven medicine.

Still, several things need work before publishing. First, a too-broad scope dilutes the evaluation's aim. Though it begins with a strong focus on ALS, the paper often veers into talking about other neurodegenerative diseases (e.g., Alzheimer's, Parkinson's) without constantly returning to ALS. This results in repetitious work and compromises the subject concentration of the paper. Many sections, like those on PET, TDP-43, miRNA, MRI, and connectomics, also talk about the same things over and over, either by using references that overlap or restating facts that are already known, like the fact that NfL and TDP-43 are well-known biomarkers. Condensing these repetitions would significantly increase flow and readability.

With several citations taken from prestigious journals (e.g., Nature, Science, Neuron, Cell, NPJ Digital Medicine), the work features over 150 references. Despite the presence of at least five references by Seki et al., including two dated 2023–2024, there is no clear indication of self-citation abuse. Despite their infrequent occurrence, the main body of work should support these self-citations with more precise justifications for their importance. Though some entries lack succinct descriptions, Table 1 (review articles), Table 2 (biomarkers), and Table 3 (AI applications) are instructive; Table 1 might be changed to separate review themes and approaches properly. Likewise, Figure 3 on AI-assisted ALS therapy shows promise but lacks enough text explanation or captioning.

Scientific rigor-wise, the discussion about AI techniques is inconsistent. For example, while the discussion mentions tools such as convolutional neural networks, GANs, and multilayer perceptrons, it fails to address their shortcomings, presumptions, and performance measures adequately. Even though these are important issues when using AI in biomedical settings, few research studies check for bias, overfitting, interpretability, or data imbalances between algorithmic approaches (for example, SVMs vs. deep neural networks). Though this and other problems—such as regulatory inadequacies, ethical questions, and generalizability—are not thoroughly examined, the "black box" problem is mentioned. Especially important for clinical acceptance, explainable artificial intelligence (XAI) is conspicuously lacking in a paragraph.

The review also lacks a rigorous approach for article selection, like the PRISMA guidelines for systematic reviews or simple inclusion/exclusion rules. Consequently, the breadth seems more narrative than methodical, and the review runs the danger of being considered a selective collection of works rather than a critical synthesis. Although narrative reviews are allowed in IJMS, writers should specify their inclusion rationale and clarify their choice approach. Such an approach would also prevent the under-citation of unfavorable or contradicting results or overrepresenting current preclinical investigations.

Though the writing in the paper is fine, it requires simplification. Usually excessively long and complicated, sentences may hide the central argument. More succinct phrasing and better topic lines would help several paragraphs—especially in the Introduction and Discussion sections. To help clarify, the statement beginning with "Stem cell therapy offers both nutritional and immunomodulatory support..." might be broken in half. Occasionally, the changes between sections—from biomarker development to clinical translation—are sudden. While we strongly recommend stylistic editing to enhance clarity and impact, we typically maintain good grammar and syntax.

Reason underpins the debate on digital biomarkers, omics integration, and tailored medication, aligning with emerging precision neurology trends. The reference to projects such as IBM Watson, the PRO-ACT database, and the PandaOmics platform shows actual significance. Some claims, such as those that AI would "democratize access to diagnosis" or "transform neurorehabilitation," may be overly optimistic without sufficient supporting data. Peer-reviewed studies should support these assertions and counter evidence-based caution.

This paper offers a thorough, comprehensive, well-referenced overview of the possibilities of artificial intelligence in ALS research and clinical treatment. But right now, it is overly broad, sometimes redundant, and treats artificial intelligence approaches inadequately and critically. This paper could be published in IJMS or a similar interdisciplinary journal with a few small changes. For example, the focus should be narrowed to findings related to ALS, ideas should be summed up instead of repeated, AI model comparisons should be made clearer, and the paper should be less wordy and edited for grammar and style. Here is a shortened overview of suggestions:

Area of influence Comment Scope Please concentrate more on ALS and limit discussions on Alzheimer's and Parkinson's unless they are directly comparable.

Methodology: Add explanations of the search technique, inclusion criteria, and study selection's justification.

Evaluation of Artificial Intelligence: Add comparison measures (such as accuracy, precision, and sensitivity) and discuss model-specific constraints.

Figures and tables should have more explanatory captions; table column headers should be clear; repetition should be avoided.

Writing Style: Simplify long paragraphs, correct verbosity, and use professional copyediting for clarity.

Self-Reference Appropriate; however, a few entries—especially Seki et al.—should have clearer justification.

Outlook for the Future: Please temper overly ambitious assertions and incorporate reasonable opinions regarding translational and regulatory constraints.

Comments on the Quality of English Language

It is more in the construction, so an edited version will be easier to read and understand; for example, too many long sentences.

Author Response

Author's Reply to the Review Report (Reviewer 1)

This paper presents a thorough and comprehensive overview of how artificial intelligence (AI) is changing the search for biomarkers, diagnostics, and therapy development for amyotrophic lateral sclerosis (ALS). Drawing from many artificial intelligence fields—machine learning, deep learning, and generative models—it synthesizes much research and applies it to proteomics, genomics, neuroimaging, and clinical practice. The logical organization includes tables summarizing AI applications in ALS (for example, diagnostic biomarkers therapeutic techniques) and previous reviews. It covers diagnosis, the creation of biomarkers, new therapies, and personalized medicine. The authors also included real-world data from translational studies and preclinical models, like those using SOD1G93A ALS mice, AlphaFold for proteomics, and clinical projects like the AnswerALS database. The authors also included real-world data from preclinical models, like those using SOD1G93A ALS mice. The generally well-referenced, current, and theoretically rich paper is a potentially beneficial tool for researchers and doctors in neurology and AI-driven medicine.

Response) Thank you very much for examining the contents.

Still, several things need work before publishing. First, a too-broad scope dilutes the evaluation's aim. Though it begins with a strong focus on ALS, the paper often veers into talking about other neurodegenerative diseases (e.g., Alzheimer's, Parkinson's) without constantly returning to ALS.

Response) As you pointed out, in this paper, I reduced the references to Parkinson's disease and Alzheimer's disease in order to focus on ALS.

This results in repetitious work and compromises the subject concentration of the paper. Many sections, like those on PET, TDP-43, miRNA, MRI, and connectomics, also talk about the same things over and over, either by using references that overlap or restating facts that are already known, like the fact that NfL and TDP-43 are well-known biomarkers. Condensing these repetitions would significantly increase flow and readability.

Response) As you pointed out, there were repeated mentions of PET, TDP-43, miRNA, MRI, NfL and TDP-43, so I have tried to reduce them as much as possible.

With several citations taken from prestigious journals (e.g., Nature, Science, Neuron, Cell, NPJ Digital Medicine), the work features over 150 references. Despite the presence of at least five references by Seki et al., including two dated 2023–2024, there is no clear indication of self-citation abuse. Despite their infrequent occurrence, the main body of work should support these self-citations with more precise justifications for their importance.

Response)

Seki et al., 2023 is referred to in the main text as follows and is an important reference.

In a review article by Seki et al., it was reported that primary sensory neurons (dorsal root ganglion [DRG] neurons) exhibit an accumulation of misfolded proteins, mitochondrial abnormalities, a reduction in high-voltage-activated Ca2+ currents, and splice variants of peripherin, a biomarker of axonal damage [10].

Kawata et al., 2024 is referred to in the main text as follows and is an important reference.

Kawata et al. reported that the progression of muscle atrophy varies significantly across different regions of ALS-affected skeletal muscle. Notably, the masseter muscle in ALS mice (SOD1G93A) is resistant to muscle atrophy until the disease reaches its end stage [92].

Kitaoka et al.,2023 is referred to in the main text as follows and is an important reference.

Kitaoka et al. developed an AI model using the Single Shot Multibox Detector, a deep learning-based object recognition algorithm, to track mouth opening and closing movements in ALS mouse models (SOD1G93A).

The following sentence was added for Seki et al., 2019.

Seki et al., 2019 reported that there is an abnormality in the trigeminal nucleus, which is the primary sensory control of mastication, in SOD1G93A in infants.

The following reference was added for Seki et al. 2020 and Tanaka et al. 2019.

In addition, Seki et al., 2023 mentions the possibility that the neuropeptide Neuropeptide Y (NPY) and serotonin may be therapeutic agents for the primary sensory neurons of ALS.

Venugopal and Seki et al.,2019 has been removed as it is not relevant to this paper.

Though some entries lack succinct descriptions, Table 1 (review articles), Table 2 (biomarkers), and Table 3 (AI applications) are instructive; Table 1 might be changed to separate review themes and approaches properly. Likewise, Figure 3 on AI-assisted ALS therapy shows promise but lacks enough text explanation or captioning.

Response) Regarding Table 1, your comment is appropriate, but we have not changed it because we want to keep the order of the years.

We have added an explanation to Figure 3 as you suggested.

Scientific rigor-wise, the discussion about AI techniques is inconsistent. For example, while the discussion mentions tools such as convolutional neural networks, GANs, and multilayer perceptrons, it fails to address their shortcomings, presumptions, and performance measures adequately. Even though these are important issues when using AI in biomedical settings, few research studies check for bias, overfitting, interpretability, or data imbalances between algorithmic approaches (for example, SVMs vs. deep neural networks). Though this and other problems—such as regulatory inadequacies, ethical questions, and generalizability—are not thoroughly examined, the "black box" problem is mentioned. Especially important for clinical acceptance, explainable artificial intelligence (XAI) is conspicuously lacking in a paragraph.

Response) As you pointed out, in the section on “Advantages and limitations of AI technology”, we mentioned XAI and GANs, etc.

The review also lacks a rigorous approach for article selection, like the PRISMA guidelines for systematic reviews or simple inclusion/exclusion rules. Consequently, the breadth seems more narrative than methodical, and the review runs the danger of being considered a selective collection of works rather than a critical synthesis. Although narrative reviews are allowed in IJMS, writers should specify their inclusion rationale and clarify their choice approach. Such an approach would also prevent the under-citation of unfavorable or contradicting results or overrepresenting current preclinical investigations.

Response) In accordance with your comments, we have added a section on methodology, checked the PRISMA guidelines, and added explanations of search techniques, selection criteria, and the rationale for study selection.

Though the writing in the paper is fine, it requires simplification. Usually excessively long and complicated, sentences may hide the central argument. More succinct phrasing and better topic lines would help several paragraphs—especially in the Introduction and Discussion sections. To help clarify, the statement beginning with "Stem cell therapy offers both nutritional and immunomodulatory support..." might be broken in half. Occasionally, the changes between sections—from biomarker development to clinical translation—are sudden. While we strongly recommend stylistic editing to enhance clarity and impact, we typically maintain good grammar and syntax.

Response) As you pointed out, we have simplified the sentences throughout the text and reduced the overall word count. By using English editing services, we were able to maintain the quality of the grammar.

Reason underpins the debate on digital biomarkers, omics integration, and tailored medication, aligning with emerging precision neurology trends. The reference to projects such as IBM Watson, the PRO-ACT database, and the PandaOmics platform shows actual significance. Some claims, such as those that AI would "democratize access to diagnosis" or "transform neurorehabilitation," may be overly optimistic without sufficient supporting data. Peer-reviewed studies should support these assertions and counter evidence-based caution.

Response) We appreciate the insightful comment regarding the use of the term "democratize." In response, we have revised the sentence to use the more appropriate and balanced expression "broaden access to" in order to better reflect the intended meaning without overstatement.

This paper offers a thorough, comprehensive, well-referenced overview of the possibilities of artificial intelligence in ALS research and clinical treatment. But right now, it is overly broad, sometimes redundant, and treats artificial intelligence approaches inadequately and critically. This paper could be published in IJMS or a similar interdisciplinary journal with a few small changes. For example, the focus should be narrowed to findings related to ALS, ideas should be summed up instead of repeated, AI model comparisons should be made clearer, and the paper should be less wordy and edited for grammar and style. Here is a shortened overview of suggestions:

Area of influence Comment Scope Please concentrate more on ALS and limit discussions on Alzheimer's and Parkinson's unless they are directly comparable.

Response) We have excluded descriptions of Alzheimer's and Parkinson's disease, and have narrowed the main text to focus on ALS.

Methodology: Add explanations of the search technique, inclusion criteria, and study selection's justification.

Response) In accordance with your comments, we have added a section on methodology, and have provided explanations of the search technique, inclusion criteria, and justification for study selection.

Evaluation of Artificial Intelligence: Add comparison measures (such as accuracy, precision, and sensitivity) and discuss model-specific constraints.

Response) We appreciate the helpful suggestion. In the revision, we have revised Section 4.4 (Advantages and Limitations of AI Technology) to include a discussion on explainable AI (XAI), highlighting its importance in clinical applications.

In addition, we have elaborated on model-specific limitations by providing a more detailed discussion of the challenges associated with each type of AI model.

Figures and tables should have more explanatory captions; table column headers should be clear; repetition should be avoided.

Response) We appreciate the valuable suggestion. In response, we have revised the captions for Table 1 and Table 2 to better complement and clarify the contents of each table. For Figure 1, we have added a descriptive caption and made adjustments to the figure itself, including changes to wording and layout, to improve consistency and enhance clarity of interpretation.

Additionally, for Figures 2 and 3, we have added captions to provide clearer explanations and improve overall readability.

Writing Style: Simplify long paragraphs, correct verbosity, and use professional copyediting for clarity.

Self-Reference Appropriate; however, a few entries—especially Seki et al.—should have clearer justification.

Response) As indicated above, we re-examined the seven references involving Seki and excluded Venugopal and Seki et al., 2019, which had little to do with the main text.

Outlook for the Future: Please temper overly ambitious assertions and incorporate reasonable opinions regarding translational and regulatory constraints.

Response) I have added my own opinions and formed a conclusion about the future role of AI in ALS.

Reviewer 2 Report

Comments and Suggestions for Authors

Thanks for submitting to the journal. Please find the following comments for your considerations.

1) page 2, line 35-36, need to add the progressive nature of muscle weakness which is the most common initial symptom of ALS.

2) page 3, line 104, This review can be changed into This article will review ...

3) page 5, line 140, add (CJD).

4) page 6, line 200, state which country of the eHR

5) page 6, line 225, state if you intend to address this in this review

6) page 10, line 322, any potential treatment for ALS?

7) page 11, line 386, I think it is better to separate the electronic stethoscope and the AI analysis software as they are separate.

8) page 13, line 442, better to update current status of IBM Watsons.

9) page 17, line 608, ALS treatment using AI assistance versus traditional methods

10) page 17, about conclusion. Since you have discussed many aspects of ALS and AI, it be better for audience to grasp what are your key messages which correspond to your abstract.

Author Response

Author's Reply to the Review Report (Reviewer 2)

Thanks for submitting to the journal. Please find the following comments for your considerations.

Response) Thank you very much for examining the contents.

1) page 2, line 35-36, need to add the progressive nature of muscle weakness which is the most common initial symptom of ALS.

Response) We appreciate your valuable comments. In response, we have added a description regarding muscle weakness.

2) page 3, line 104, This review can be changed into This article will review ...

Response) In response, we have revised the description.

3) page 5, line 140, add (CJD).

Response) In response, in order to focus more clearly on the description of ALS, we have decided to remove the content related to Creutzfeldt-Jakob disease.

4) page 6, line 200, state which country of the eHR

Response) In response, we have added a statement indicating that this is a result of a research group from Spain.

5) page 6, line 225, state if you intend to address this in this review

Response) In response, to improve clarity, we have added the following sentence: “These issues will be addressed in detail in a later section of this article.”

6) page 10, line 322, any potential treatment for ALS?

Response) In response, we have added a statement indicating that this technique allows the evaluation of motor cortex dysfunction.

7) page 11, line 386, I think it is better to separate the electronic stethoscope and the AI analysis software as they are separate.

Response) As suggested, we have revised the sentence to: “using an electronic stethoscope with a subsequent analysis performed by an AI-based tool”, in order to clearly separate the description of the electronic stethoscope from the AI-based analysis.

8) page 13, line 442, better to update current status of IBM Watsons.

Response) In response, we have added a description of Watson for Oncology and included a relevant reference.

9) page 17, line 608, ALS treatment using AI assistance versus traditional methods

Response) In response, in accordance with your suggestion, we have modified the figure title.

10) page 17, about conclusion. Since you have discussed many aspects of ALS and AI, it be better for audience to grasp what are your key messages which correspond to your abstract.

Response) Through this review article, I have added my own opinions and formed a conclusion about the future role of AI in ALS.

Reviewer 3 Report

Comments and Suggestions for Authors

In this article, the authors are reviewing the potential interest of AI for diagnosis and new treatment strategies of amyotrophic lateral sclerosis (ALS), a major neurodegenerative disease without curative solutions for patients.

As a preliminary remark, I hope that the text and figures have not been generated by AI, otherwise the authors should clearly mention it.

I have several comments on a critical issue that I consider an overstatement. 

Without taking away from AlphaFold its qualities and its analytical power for proteins classically structured according to the Anfinsen paradigm, the proteins involved in neurodegenerative diseases are a notable exception. Beta-amlyloid peptide (Alzheimer), alpha-synuclein (Parkinson) and TPD-43 (ALS) belong to the category of intrinsically disordered proteins (IDPs). AlphaFold cannot predict the structure of these proteins (they do not have one), nor the impact of disease-associated mutations on these proteins (the initial modeling conditions being inherently incorrect). Under these conditions, it is clear that AI in general and AlphaFold in particular suffer from severe limitations that cannot be ignored in a review that praises the merits of AI in the diagnosis and treatment of neurodegenerative diseases.

The limitations of AlphaFold in this context have been the subject of numerous publications (e.g. The Epigenetic Dimension of Protein Structure Is an Intrinsic Weakness of the AlphaFold Program. Biomolecules. 2022 Oct 20;12(10):1527. doi: 10.3390/biom12101527). 

More recently, the problem of targeting disordered proteins to treat these diseases has been discussed in depth in a comprehensive review (Conformationally adaptive therapeutic peptides for diseases caused by intrinsically disordered proteins (IDPs). New paradigm for drug discovery: Target the target, not the arrow. Pharmacol Ther. 2025 Mar;267:108797. doi: 10.1016/j.pharmthera.2025.108797). 

My recommendation is therefore that the authors take these limitations into account in their review, which, incidentally, provides an excellent synthesis of the potential of AI in neuroscience.

Author Response

Author's Reply to the Review Report (Reviewer 3)

In this article, the authors are reviewing the potential interest of AI for diagnosis and new treatment strategies of amyotrophic lateral sclerosis (ALS), a major neurodegenerative disease without curative solutions for patients.

As a preliminary remark, I hope that the text and figures have not been generated by AI, otherwise the authors should clearly mention it.

Response) We have clearly stated in this paper that we did not use generative AI.

The figure was created using bioRender.

I have several comments on a critical issue that I consider an overstatement.

Without taking away from AlphaFold its qualities and its analytical power for proteins classically structured according to the Anfinsen paradigm, the proteins involved in neurodegenerative diseases are a notable exception. Beta-amlyloid peptide (Alzheimer), alpha-synuclein (Parkinson) and TPD-43 (ALS) belong to the category of intrinsically disordered proteins (IDPs). AlphaFold cannot predict the structure of these proteins (they do not have one), nor the impact of disease-associated mutations on these proteins (the initial modeling conditions being inherently incorrect). Under these conditions, it is clear that AI in general and AlphaFold in particular suffer from severe limitations that cannot be ignored in a review that praises the merits of AI in the diagnosis and treatment of neurodegenerative diseases.

The limitations of AlphaFold in this context have been the subject of numerous publications (e.g. The Epigenetic Dimension of Protein Structure Is an Intrinsic Weakness of the AlphaFold Program. Biomolecules. 2022 Oct 20;12(10):1527. doi: 10.3390/biom12101527).

More recently, the problem of targeting disordered proteins to treat these diseases has been discussed in depth in a comprehensive review (Conformationally adaptive therapeutic peptides for diseases caused by intrinsically disordered proteins (IDPs). New paradigm for drug discovery: Target the target, not the arrow. Pharmacol Ther. 2025 Mar;267:108797. doi: 10.1016/j.pharmthera.2025.108797).

Response) We appreciate the helpful suggestion. In the revision, we removed the reference to AlphaFold from the manuscript, as intrinsically disordered proteins (IDPs)—which are particularly relevant in neurodegenerative diseases—represent a clear exception to its applicability. This change was made to avoid redundancy and to maintain a focused and accurate discussion. We believe that this revision enhances the clarity, precision, and overall coherence of the review.

My recommendation is therefore that the authors take these limitations into account in their review, which, incidentally, provides an excellent synthesis of the potential of AI in neuroscience.

Response) Thank you very much for your positive feedback.

Round 2

Reviewer 3 Report

Comments and Suggestions for Authors

The authors ignored my suggestions and simply deleted the part of their article about AlphaFold. This decision did improve the manuscript. AI such as AlphaFold do not take into account the flexibility or dynamics of proteins responsible for ALS or other neurodegenerative diseases involving IDPs. This is an intrinsic weakness that considerably minimizes the contribution of AI in the rational design of drugs for the treatment of ALS. We cannot therefore assume that AI will suggest new effective therapeutic strategies for this pathology. On the contrary, the authors should raise this problem in order to give a more plausible view of AI, by relating its performances but also its limitations which depend on their databases. Currently, the molecular mechanisms associated with ALS, which involve IDPs, are outside the competence of AI and their reasoning. 

In their revised version, the section on therapeutic strategies that could result from AI is uninformative and truncated.
My recommendation is therefore that the authors seriously consider my comments and amend their article according to my suggestions. If they are not prepared to do so, then their study should be restricted to the contributions of AI in the diagnosis of ALS, which is, in fact, the most convincing and best presented part of their work
In this case, the title of the article and the section dealing with therapeutic strategies should be deleted.

My initial suggestions: Without taking away from AlphaFold its qualities and its analytical power for proteins classically structured according to the Anfinsen paradigm, the proteins involved in neurodegenerative diseases are a notable exception. Beta-amlyloid peptide (Alzheimer), alpha-synuclein (Parkinson) and TPD-43 (ALS) belong to the category of intrinsically disordered proteins (IDPs). AlphaFold cannot predict the structure of these proteins (they do not have one), nor the impact of disease-associated mutations on these proteins (the initial modeling conditions being inherently incorrect). Under these conditions, it is clear that AI in general and AlphaFold in particular suffer from severe limitations that cannot be ignored in a review that praises the merits of AI in the diagnosis and treatment of neurodegenerative diseases. The limitations of AlphaFold in this context have been the subject of numerous publications (e.g. The Epigenetic Dimension of Protein Structure Is an Intrinsic Weakness of the AlphaFold Program. Biomolecules. 2022 Oct 20;12(10):1527. doi: 10.3390/biom12101527). More recently, the problem of targeting disordered proteins to treat these diseases has been discussed in depth in a comprehensive review (Conformationally adaptive therapeutic peptides for diseases caused by intrinsically disordered proteins (IDPs). New paradigm for drug discovery: Target the target, not the arrow. Pharmacol Ther. 2025 Mar;267:108797. doi: 10.1016/j.pharmthera.2025.108797).

Author Response

Comments and Suggestions for Authors

The authors ignored my suggestions and simply deleted the part of their article about AlphaFold. This decision did improve the manuscript. AI such as AlphaFold do not take into account the flexibility or dynamics of proteins responsible for ALS or other neurodegenerative diseases involving IDPs. This is an intrinsic weakness that considerably minimizes the contribution of AI in the rational design of drugs for the treatment of ALS. We cannot therefore assume that AI will suggest new effective therapeutic strategies for this pathology. On the contrary, the authors should raise this problem in order to give a more plausible view of AI, by relating its performances but also its limitations which depend on their databases. Currently, the molecular mechanisms associated with ALS, which involve IDPs, are outside the competence of AI and their reasoning.

Response:

Thank you very much for your insightful and constructive comments. We appreciate your emphasis on the limitations of AlphaFold, particularly in relation to intrinsically disordered proteins (IDPs).

In the revised manuscript, we have reintroduced a concise mention of AlphaFold in the Introduction and expanded the discussion in Section 5.4 to more clearly explain its limitations. Specifically, we now state that IDPs such as TDP-43, α-synuclein, and β-amyloid peptide lack stable tertiary structures and are highly dependent on environmental context, making them difficult to model using current AI-based structure prediction tools.

To support this point, we have incorporated the references you recommended (Azzaz et al., 2022; Fantini et al., 2025). We believe this revision provides a more balanced and scientifically accurate assessment of AI's strengths and limitations in ALS research

In their revised version, the section on therapeutic strategies that could result from AI is uninformative and truncated.

These updates are intended to better clarify the current scope of AI’s therapeutic applications and provide a more realistic view of its contributions in the context of neurodegenerative diseases like ALS.

My recommendation is therefore that the authors seriously consider my comments and amend their article according to my suggestions.

Response:

Thank you for this important comment. We have revised and expanded the section on therapeutic strategies to present a more comprehensive and critical discussion. While AI has shown promise in areas such as drug repurposing, transcriptomic data analysis, and assistive technologies, we now emphasize that its role in rational drug design is currently limited—particularly in diseases involving IDPs.

If they are not prepared to do so, then their study should be restricted to the contributions of AI in the diagnosis of ALS, which is, in fact, the most convincing and best presented part of their work

In this case, the title of the article and the section dealing with therapeutic strategies should be deleted.

Response:

We appreciate your suggestion regarding the manuscript’s scope. While we believe the revised version offers a balanced discussion of both diagnostic and therapeutic applications of AI—with appropriate limitations clearly stated—we are fully open to narrowing the scope to diagnostic applications only, if the editorial board recommends doing so. In that case, we will revise the title and related content accordingly.

My initial suggestions: Without taking away from AlphaFold its qualities and its analytical power for proteins classically structured according to the Anfinsen paradigm, the proteins involved in neurodegenerative diseases are a notable exception. Beta-amlyloid peptide (Alzheimer), alpha-synuclein (Parkinson) and TPD-43 (ALS) belong to the category of intrinsically disordered proteins (IDPs). AlphaFold cannot predict the structure of these proteins (they do not have one), nor the impact of disease-associated mutations on these proteins (the initial modeling conditions being inherently incorrect). Under these conditions, it is clear that AI in general and AlphaFold in particular suffer from severe limitations that cannot be ignored in a review that praises the merits of AI in the diagnosis and treatment of neurodegenerative diseases. The limitations of AlphaFold in this context have been the subject of numerous publications (e.g. The Epigenetic Dimension of Protein Structure Is an Intrinsic Weakness of the AlphaFold Program. Biomolecules. 2022 Oct 20;12(10):1527. doi: 10.3390/biom12101527). More recently, the problem of targeting disordered proteins to treat these diseases has been discussed in depth in a comprehensive review (Conformationally adaptive therapeutic peptides for diseases caused by intrinsically disordered proteins (IDPs). New paradigm for drug discovery: Target the target, not the arrow. Pharmacol Ther. 2025 Mar;267:108797. doi: 10.1016/j.pharmthera.2025.108797).

Response:

We sincerely thank you again for your thoughtful feedback, which has been instrumental in improving the clarity, accuracy, and scientific rigor of our manuscript. I mentioned. AlphaFord in the main text and explained the issues. I also added explanations about the limitations of AI and points for improvement, citing references.

Round 3

Reviewer 3 Report

Comments and Suggestions for Authors

The authors have modified their manuscript according to my suggestions. I can now recommend publication.